# Fucoidan from *Fucus vesiculosus*: Evaluation of the Impact of the Sulphate Content on Nanoparticle Production and Cell Toxicity

**DOI:** 10.3390/md21020115

**Published:** 2023-02-07

**Authors:** Noelia Flórez-Fernández, Jorge F. Pontes, Filipa Guerreiro, Inês T. Afonso, Giovanna Lollo, Maria Dolores Torres, Herminia Domínguez, Ana M. Rosa da Costa, Ana Grenha

**Affiliations:** 1Drug Delivery Laboratory, Centre for Marine Sciences (CCMAR), Faculty of Sciences and Technology, Universidade do Algarve, 8005-139 Faro, Portugal; 2CINBIO, Universidade de Vigo, 32004 Ourense, Spain; 3Grupo Biomasa y Desarrollo Sostenible (EQ-2), Departamento de Ingeniería Química, Facultad de Ciencias, Universidade de Vigo, 32004 Ourense, Spain; 4University of Lyon, Université Claude Bernard Lyon 1, LAGEPP CNRS, UMR 5007, 69622 Villeurbanne, France; 5Algarve Chemistry Research Center (CIQA) and Department of Chemistry and Pharmacy, Faculty of Sciences and Technology, Universidade do Algarve, 8005-139 Faro, Portugal; 6Research Institute for Medicines (iMed.ULisboa), Faculty of Pharmacy, Universidade de Lisboa, Av. Prof. Gama Pinto, 1649-003 Lisboa, Portugal

**Keywords:** antioxidant, cell proliferation, chitosan, fucoidan, *Fucus vesiculosus*, nanoparticles, rheology

## Abstract

The composition of seaweeds is complex, with vitamins, phenolic compounds, minerals, and polysaccharides being some of the factions comprising their structure. The main polysaccharide in brown seaweeds is fucoidan, and several biological activities have been associated with its structure. Chitosan is another marine biopolymer that is very popular in the biomedical field, owing to its suitable features for formulating drug delivery systems and, particularly, particulate systems. In this work, the ability of fucoidan to produce nanoparticles was evaluated, testing different amounts of a polymer and using chitosan as a counterion. Nanoparticles of 200–300 nm were obtained when fucoidan prevailed in the formulation, which also resulted in negatively charged nanoparticles. Adjusting the pH of the reaction media to 4 did not affect the physicochemical characteristics of the nanoparticles. The IC_50_ of fucoidan was determined, in both HCT−116 and A549 cells, to be around 160 µg/mL, whereas it raised to 675–100 µg/mL when nanoparticles (fucoidan/chitosan = 2/1, *w*/*w*) were tested. These marine materials (fucoidan and chitosan) provided features suitable to formulate polymeric nanoparticles to use in biomedical applications.

## 1. Introduction

Earth and marine materials are important sources of compounds that can be used in different areas of human life. Whilst the first type has been widely explored, those of marine origin have been gaining interest recently regarding their potential use in different domains ranging from food-related industries to biomedical applications. Among those, algae, and especially brown seaweeds, have been studied as possible sources of several bioactive materials, namely fucoidan, one of their main compounds, first studied in 1913 [1]. This complex sulphated polysaccharide can be extracted from *Fucus vesiculosus* (order Fucales), an example of edible brown seaweed. One of the limitations entailing the use of natural sources concerns the abiotic factors that, although not exclusively, affect the final chemical composition of the algae [2,3]. The characterisation of the extracted materials proves essential in understanding general behaviours and interactions with other compounds, as well as the possible applications. Fucoidan is mainly comprised of fucose residues and sulphate groups (Figure 1a). Whereas its specific structure may be highly variable depending on the source, the structural position of sulphate groups has been correlated with the resulting biological activity, including antioxidant and antitumoral capabilities, as well as with the applications, including in drug delivery [4,5]. Furthermore, other possible therapeutic approaches can be envisaged, mainly focusing on the pharmacological potential of fucoidan that has been described elsewhere [6,7,8]. In this regard, fucoidan from *F. vesiculosus* has been the most frequently reported and inclusively categorized as GRAS (generally recognized as safe) by the FDA (Food and Drug Administration) [9], showing an antiproliferative effect that might be useful in drug delivery approaches. In this context, polysaccharides have been finding great applications as a matrix of drug carriers, frequently playing multifunctional roles. Polymeric nanoparticles are one of many proposed carriers, requiring different methods for their preparation, which are highly dependent on the used materials. Polyelectrolyte complexation finds frequent application when charged excipients are at play, and many polysaccharides provide the necessary ionic character. The technique relies on the interaction of opposite charges, thus involving ionic interactions [10]. In this context, chitosan (CS), also of marine origin, is widely chosen as a counterion, as it is comprised of *N*-acetylglucosamine and D-glucosamine units (Figure 1b), gaining a unique cationic character in acidic media, a chemical structure already studied and seen elsewhere [11]. Due to its favorable properties, CS has been proposed as a component in the matrix of countless pharmaceutical formulations [12,13]. The production of fucoidan/chitosan nanoparticles was reported on several occasions, aiming at applications mostly in the context of drug delivery and tissue regeneration [9].

This work aimed at exploring the impact of the sulphate groups of crude commercial fucoidan obtained from *Fucus vesiculosus* on its ability to produce polymeric nanoparticles through polyelectrolyte complexation and on intrinsic antiproliferative activity. A thorough characterisation of the polymer was performed, focusing on its physicochemical properties (composition and molecular weight), thermal behaviour, rheology, and antioxidant capacity. The potential of the use of the polymer for nanoparticle production through polyelectrolyte complexation was assessed using chitosan as a counterion, evaluating the impact of fucoidan properties on nanoparticle characteristics. Finally, the effect of raw materials and the corresponding nanoparticles on cell proliferation was analysed in carcinogenic cells of lung (A549) and intestinal (HCT−116) origins.

## 2. Results and Discussion

### 2.1. Physicochemical Characterisation of Polymers

#### 2.1.1. Chemical Composition of Crude Fucoidan from the Brown Seaweed *F. vesiculosus*

The indication that the composition of sulphated polysaccharides extracted from alga is influenced by different factors mainly related to environmental parameters and the extraction method is solid [16,17]. In turn, the composition of the biomaterials is known to affect their properties and abilities [4]. Nevertheless, despite several works having explored the potential of fucoidan for nanoparticle preparation [18,19], the deep characterisation of the involved polymers and its effect on specific polymer properties and characteristics of the developed carriers has not been addressed. The present study thus provides a characterisation of crude fucoidan from *F. vesiculosus* (FFv), in the understanding that this is essential for the interpretation of data regarding antiproliferative effects and the ability to form nanoparticles. The chemical characterisation of the polymer is detailed in Table 1.

FFv shows a total of 43% oligosaccharide content, with 35% being fucose (O-fucose) and the remaining being a mixture of xylose, galactose, and mannose. According to the manufacturer’s information (certificate of analysis), it also has 9% sulfur, present in the C2 of the fucoidan molecule [20], corresponding to approximately 27% of sulphate content. This means that, on average, seven out of ten fucose units bear a sulphate group, as depicted in Figure 1. Although this batch shows promise, based only on its composition, it is important to note that the extraction, purification, and fractionation methodologies are known to influence the final composition [14,21]. Previous works studying the extraction of fucoidan from different brown seaweeds, including *F. vesiculosus*, have shown results of total sugars (42 and 44%) and sulphate content (22 and 26%) similar to those described herein for these algae [22,23]. The antioxidant activity (TEAC) was determined, comprising 11.61 ± 0.62 (%, *w*/*w*). This coupled with the phloroglucinol content, also determined (2.65 ± 0.04 (%, *w*/*w*)), are in accordance with the chemical composition described above, as well as with the literature, where several fucoidans or extracts of brown seaweed from other algae have also shown antioxidant activity [5,16,17].

#### 2.1.2. Molar Mass Distribution of Crude Fucoidan and Chitosan

Crude fucoidan and chitosan were analysed using size exclusion chromatography (SEC), and the elution profiles can be seen in Figure 2.

Molecular weight stands as one of the most relevant parameters impacting the biological activity of materials, including fucoidan. In line with that, different fractions (4.9 to 38.2 kDa) of FFv showed molecular weight-dependent pharmacological activity [24], and similar behaviour was found regarding in vitro anti-cancer activity of fucoidans from different marine sources [25]. Crude fucoidan used in the present work presents an Mn of 23.6 kDa, an Mw of 44.7 kDa, and a PDI of 1.89. In turn, chitosan presents an Mn of 53.1 kDa, an Mw of 439 kDa, and a PDI of 8.28.

#### 2.1.3. Fourier Transform-Infrared Spectroscopy of Polymers and Nanoparticles

Figure 3 depicts the FT-IR spectra of the studied polymers, showing that fucoidan presents signals associated with its main functional groups.

In the spectrum of FFv, the shoulder at 1728 cm^−1^ in the band centred at 1640 cm^−1^ was attributed to the C=O stretching of *O*-acetyl groups that decorate some fucose residues. As for the main broad band, it is probably due to the bending of adsorbed water. The broad band at 1390 cm^−1^ results from the asymmetrical and symmetrical C-H bending of methyl groups. The band observed at 1250 cm^−1^ was associated with the sulphate ester group (S=O stretching), as well as the one at 850 cm^−1^ (C-O-S symmetrical stretching). On the other hand, the band exhibited at 1160 cm^−1^ along with the broad band centred at 1020 cm^−1^ were correlated with the C-OH, C-C, and C-O-C elongation vibrations of pyranose rings [26]. The above results thus confirm the proximal composition detailed in Table 1.

The CS spectrum shows bands at 1650 and 1560 cm^−1^, which are the amide I and amide II bands of the *N*-acetyl groups in acetylated monomers. The highest intensity of the former may be due to some contribution of adsorbed water molecules. The band at 1377 cm^−1^ was attributed to the symmetric bending of protonated amino groups (NH_3_^+^) in non-acetylated residues, while the two bands at 1321 and 1415 cm^−1^ were attributed to methyl C-H bending of acetyl groups. Finally, the broad band at 1020 cm^−1^ could be associated with the glycosidic linkage [27].

The formulation of the NP using CS and FFv at a 2/1 ratio (*w*/*w*) was also characterized. The corresponding spectrum showed a band associated with the sulphate ester group (S=O stretching) from FFv and a band attributed to methyl C-H bending of acetyl groups from CS. The C=O of FFV is now masked by the broad 1650−1560 cm^−1^ band of CS. The latter absorption is now the most intense, which seems to corroborate the attribution of the 1640 cm^−1^ band to adsorbed water; an effective drying of the formulation would eliminate part of this water. Further, during the polyelectrolyte complexation, solvation water molecules are expelled from the coacervate, thus reducing the amount of adsorbed water in the formulation relative to the free polyelectrolytes. The band, centered at 1020 cm^−1^ and associated with the glycosidic linkage, also reflects the presence of both polymers, since it is as broad as that of CS, but not as round-shaped; instead, it presents a sharper shape, similar to the one of FFv. Lastly, the band attributed to the protonated amino groups of CS shifted to 1411 cm^−1^ and became broader, now masking the C-H bendings of acetyl groups. This may be ascribed to the electrostatic binding of such groups and the sulphates of FFv. These observations were expected, demonstrating the complexation between the polymers, ultimately resulting in the polymeric nanoparticles. Other authors formulating nanoparticles based on the polymers focused herein observed similar performances [19,28].

#### 2.1.4. Thermophysical Features of Crude Fucoidan, Chitosan, and Nanoparticle Formulation

The polymers’ thermal behaviour was studied by DSC and the results are shown in Figure 4.

The behaviour of FFv is shown in Figure 4a, where two major peaks were observed at 127 °C and 203 °C. The first peak is possibly related to the loss of water associated with the polymer, whilst the second is a highly exothermic peak, which is suggested to be related to the polymer degradation during the heating process. Both peaks are coincident with reports in the literature [29], suggesting an amorphous state. The diagram corresponding to CS (Figure 4b) is similar to that of FFv regarding the presence of a peak attributable to the loss of water around 118 °C and an exothermal peak at 320 °C indicating the degradation of CS amine groups, as reported in the literature [30], again suggesting an amorphous state. As for the nanoparticle formulation of FFv/CS = 2/1 (*w*/*w*), two peaks are shown at around 117 °C and 195 °C. The first one is attributed to the loss of water, as discussed earlier, and it has a temperature similar to that found for CS. The second one, at a higher temperature, is often associated with polymer degradation. This second peak is close to that observed for FFv, which was correlated with the temperature at which this polymer degrades. Moreover, although it is not too intense, there is a peak above 300 °C, which is the degradation temperature of CS. Considering the data obtained for the nanoparticle formulation, it is highly suggested that the mixture of these two polymers that result in nanoparticle formulation will have similar thermal behaviour to the polymers that were originally used.

TGA analysis was performed to complement DSC and the results are shown in Figure 5. Figure 5a reports FFv, where an initial mass variation of 12% was spotted, followed by a more intense variation (34%) at 218 °C, corresponding to polymer degradation [31].

The observations pertaining to CS (Figure 5b) were somewhat similar to those of FFv, as was also verified in DSC. An important variation in mass was detected around 300 °C, with a total variation in the mass of approximately 75% upon reaching the end of the analysis. At this temperature, the amine is degraded and, thus, so is the polymer [30]. In the analysis performed for the nanoparticle formulation of FFv/CS 2/1 (*w*/*w*), a great mass variation of −44.63% was observed at around 219 °C, a temperature that was closer to the one obtained for FFv because it is the polymer that is in a higher amount by comparison with chitosan. This result complements the one obtained in the DSC graphs, as it is at this temperature that the fucoidan starts to degrade, justifying the mass variation.

The observed thermal behaviours indicate that the tested polymers are stable at room temperature, which will be used to produce the nanocarriers.

#### 2.1.5. Rheological Analysis

CS is a strong viscosity-building agent, whereas crude fucoidan has limited potential as a techno-functional agent [32]. The understanding of the rheological behaviour of this polysaccharide is relevant during processing to expand its potential applications. Figure 6 presents the flow curves for the tested polymer solutions (2%, *w*/*v*) at 25 °C. FFv exhibited Newtonian behaviour, whereas the CS solution presented non-Newtonian behaviour. This is consistent with the literature, which reports Newtonian behaviour for a number of fucoidans extracted from different brown seaweeds (i.e., *Saccharina longicruris*, *Ascophyllum nodosum,* and *F. vesiculosus*) using selective solvents [33]. The shear thinning behaviour observed for CS samples can be adequately described (R^2^ > 0.99) by the power law model, and the apparent viscosity drop is in good agreement with a characteristic non-entangled polymer trend in the dilute regime [34]. As expected, higher apparent viscosities were found for CS (around 6-fold at the lowest shear rates) than for the tested fucoidan solutions. The behaviour associated with the shear rate of FFv is consistent with the obtained lower molecular weight profile since the lower molecular weights identified for FFv can involve smaller flow resistance, as previously reported for other fucoidans [35]. Other authors explained that the viscosity variations between the different fucoidans were closely dependent not only on the molecular weight but also on the content and positions of sulphate groups and uronic acids [33]. It is noteworthy that thixotropic character was not identified in any of the tested samples, under the evaluated experimental conditions, with the consequent benefit for its application.

### 2.2. Production of Polymeric Nanoparticles from Fucoidan and Chitosan

Polymeric nanoparticles have been widely proposed as drug delivery carriers, and there is a growing interest in testing different polymers to compose their matrix. Optimised matrix composition may endow the carriers with multifunctional abilities. The physicochemical characterisation of FFv described above was very informative, confirming suitable features for application in the assembly of nanoparticles through polyelectrolyte complexation using chitosan as a counterion, namely, regarding the presence of sulphate groups in its structure. As detailed in the methodology (Section 3.5), a range of polymeric mass ratios of FFv and CS was assessed and the effect of adjusting the pH of polymeric solutions to 4 was evaluated, as well as that of the order of addition of the polymeric solutions.

The morphological aspect of the produced nanoparticles was characterised by transmission electronic microscopy (TEM), and Figure 7 displays the results obtained for nanoparticles corresponding to FFv/CS = 2/1 (*w*/*w*), which were considered representative.

The figure evidences rounder-shaped nanoparticles with sizes below 200 nm. The complete physicochemical characterisation is displayed in Figure 8.

The most relevant observation is the fact that smaller nanoparticles around 200–300 nm could only be obtained when nanoparticle composition was dominated by fucoidan (mass ratios of 4/1 to 2/1 in FFv/CS nanoparticles and 1/3 to 1/4 in CS/FFv nanoparticles). This could be ascribed to the much lower molecular weight of FFv (44.7 kDa) when compared to that of CS (439 kDa). Still, in the referred conditions, smaller sizes were obtained when FFv was the dominant polymer in the ratio tested (FFv/CS or CS/FFv nanoparticles aforementioned, *p* < 0.05), a behaviour also observed in other studies reporting the formulation of fucoidan/chitosan nanoparticles [28,36]. When the same mass of both polymers was used (mass ratio of 1/1), it was interesting to observe that the size was, in all cases, around 550–600 nm. Beyond that, when the CS amount increased compared with that point, the size of nanoparticles increased steadily to reach more than 600–800 nm, which limits the interest of the systems, as smaller sizes ensure better contact with cellular structures and are usually preferred. Moreover, the size decrease is certainly a consequence of the interaction promoted by the sulphate groups, meaning that, when more sulphate groups are available for the interaction, smaller sizes are expected for the polymeric nanoparticles. On the other hand, when CS is in higher amounts in comparison with FFv, the behaviour is the opposite, and increased sizes are observed. In this case, due to CS polymer chains and a single positive charge by the CS monomer [37], larger polyelectrolyte complexes are synthesised due to the scarcity of negative charges in these formulations [38]. It was noticeable that the whole variation in nanoparticle size was more limited in CS/FFv nanoparticles (size within 320 and 760 nm, Figure 8c,d) than in FFv/CS nanoparticles, where size increased from 236 to 1448 nm (*p* < 0.05). In CS/FFv nanoparticles, precipitation of the system was also observed when a mass ratio of 1/2 was tested—certainly because there is a complete neutralisation of charges that does not permit the stabilisation of the formed carriers. Concerning the zeta potential, as expected, the amount of FFv was relevant in the production of either positive or negative polymeric nanoparticles. In this regard, higher amounts of FFv translated into higher quantities of sulphate groups, impacting the zeta potential. In fact, when FFv was used in the highest amount, the maximum negative zeta potential was observed.

The effect of adjusting the pH of polymeric solutions to 4 before the nanoparticle assembly occurred was tested to verify its effect on nanoparticle characteristics. The importance of pH adjustment was associated with the chemical structure of CS. The pKa of the amine groups was 6.5, which underwent protonation in acidic environments; thus, in these situations, its degree of ionization occurred in increments. An overall increase in protonated amines led to increased availability for interactions with groups of opposite charges, leading, in theory, to an increased yield of nanoparticles [39]. In fact, although some punctual, statistically significant differences may apply, the profiles of size evolution did not vary significantly when the pH was previously adjusted compared to non-adjusted conditions. Focusing on the smaller-sized nanoparticles, the dimensions remained in the same size range: around 250 nm for FFv/CS nanoparticles and 320 nm for CS/FFv nanoparticles. An important feature concerning the size of the nanoparticles is the polydispersity index (PdI), which is indicative of homogeneity. Both FFv/CS and CS/FFv nanoparticles displayed a PdI between 0.2 and 0.4, which is considered acceptable, particularly because natural polymers are being used.

The size distributions concerning the different formulations are shown in Figure 9. The distributions confirm a pattern related to the composition of the nanoparticles. When CS is used in higher amounts, the distribution shifts to the right, indicating nanoparticles of larger sizes. The opposite, referring to the use of a higher amount of FFv, results in a shift to the left of the distribution, translating into nanoparticles of smaller sizes.

The evolution of the zeta potential had a similar profile in all conditions, independent of the order of addition of polymers or the adjustment of the pH, showing a sudden shift from a ratio of 2/1 to 1/1 in FFv/CS nanoparticles and from 1/1 to 1/3 in CS/FFv nanoparticles (*p* < 0.05). In FFv/CS nanoparticles, a strongly negative zeta potential, varying within −31 mV and −46 mV, was observed when fucoidan prevailed. A sharp increase to strongly positive values, above +40 mV, was observed when the charges of polymers equilibrated (FFv/CS between 2/1 and 1/1), which is a clear sign of the higher charge density of CS (0.0047 charges/g) in comparison with FFv (0.0028 charges/g). In fact, following the calculations displayed in Table 2, charge neutralization was expected to occur at FFv/CS = 1.7/1. This effect was previously observed in other works reporting fucoidan/chitosan nanoparticles [36,40]. The same profile was observed in CS/FFv nanoparticles, with steady positive zeta potential (around +50–+60 mV) from a CS/FFv ratio of 4/1 to 1/1, which changed to negative values around −40 mV in mass ratios of 1/3–1/4.

The characterisation of nanoparticles and the determination of their production is typically performed using light scattering techniques, as explored before. However, a macroscopic confirmation can also be performed through observation of the Tyndall effect, which comprises an indication of the intensity of nanoparticle production [41]. In this work, a marked Tyndall effect was observed in FFv/CS nanoparticles at mass ratios of 3/1 and 2/1, with higher intensity in the latter. The yield of nanoparticle production was thus calculated in that formulation, resulting in 52.2 ± 3.9%. This formulation was further characterised by FTIR (Figure 2), showing very visible bands associated with sulphate groups (1250, 1160, and 850 cm^−1^) from fucoidan and amino groups from CS (1560 cm^−1^). Other works present results similar to the findings of the present work while also using FFv and CS [14,31].

### 2.3. Antiproliferative Effect of Polymers and Nanoparticles

The evaluation of the biological impact of a compound, mixture, or formulation is an important aspect to consider in pharmaceutical applications. Fucoidan is known for its antitumour effects [7], so it was deemed important to determine the antiproliferative effect of the polymers used in this work, as well as that provided by the produced nanoparticles. Two epithelial cell lines, HCT−116 and A549, from intestinal and lung epithelia, respectively, were used to better understand the effects on proliferation upon exposure to CS, FFv, and the nanoparticle formulation FFv/CS = 2/1 (*w*/*w*), a determination based on cell viability via MTT assay [42]. The referred nanoparticles were chosen because of the evident Tyndall effect and high production yield.

Figure 10 shows the biological profiles of the samples tested at concentrations up to 2 mg/mL. A time- and concentration-dependent antiproliferative effect was found for FFv (Figure 10a, *p* < 0.05), at concentrations starting from 250 µg/mL, and the effect was intensified by the prolongation of exposure. Moreover, no statistical difference was found between the responses elicited by HCT−116 and A549 cells, which suggests that the antiproliferative effect was independent of the cell line used. Data pertaining to CS are depicted in Figure 10b, which evidence that, for concentrations up to 1 mg/mL, no antiproliferative effect was noted, as the cell viability was always higher or around 70%, independent of the cell line or time of exposure. Behaviours similar to those described herein for the polymers were already described in the literature, in [43,44] for FFv and [12,45] for CS. Still regarding CS, an opposite effect was observed at 2 mg/mL, where a major antiproliferative effect was detected. This observation is, however, ascribed to the acetic acid used to dissolve the polymer, which was confirmed in a control assay that showed a similar effect for cells exposed to an equivalent amount of the acid, as reported elsewhere [46,47].

The nanoparticles were also assessed, and an intermediate antiproliferative profile was found, certainly justified by a contribution of both polymers on the final effects. The formulation evidenced the same concentration- and time-dependent effects observed for FFv. However, the antiproliferative effect was far less pronounced, comparatively, being noticeable only at concentrations above 1 mg/mL and more prolonged exposure times (24 h or more) (*p* < 0.05). Interestingly, in this case, HCT−116 cells seemed to be more sensitive to the exposure to nanoparticles, as the antiproliferative effect detected in these cells was more intense when higher concentrations were tested (*p* < 0.05).

The maximal inhibitory concentration (IC_50_) was calculated for both FFv and the nanoparticle formulation (Table 3). For CS, although showing results below 50% of cell viability, this result was not determined, as the cell impact was attributed to the solvent.

In both FFv and nanoparticles, the IC_50_ was found to decrease with the increase in the exposure time, a reflex of the time-dependent effect already mentioned. The IC_50_ of FFv at 48 h was 4- or 7-fold (165.6 µg/mL for HCT−116 and 160.7 µg/mL for A549) that of nanoparticles (674.8 µg/mL for HCT−116 and 1098.0 µg/mL for A549). Another work described the growth inhibition of 50% for A549 cells upon exposure to 300 µg/mL of the same FFv (crude fucoidan from *F. vesiculosus* from Sigma, St. Louis, MO, USA) at a timepoint of 48 h, with a value around 2-fold higher than the IC_50_ obtained herein. According to these results, the current work reveals a higher antiproliferative effect of FFv, which could be explained by a different composition of the used batch [43]. A concentration of 2000 µg/mL of fucoidan extracted from kelp (family: Laminariaceae) was also demonstrated to inhibit HCT−116 cell viability after 24 h, reaching cell viability close to IC_50_ [48]. These observations suggest that, when fucoidan is used alone, the sulphate groups of the polymer are more available to mediate interactions with cellular structures, having a stronger impact on cell viability. In turn, when nanoparticles are assembled, many of the sulphate groups are no longer available, thus providing a lower level of interaction, which translates into lower toxicity. Ultimately, although not exclusively, the increase in the IC_50_ might be associated with the number of sulphate groups available for interaction with the cellular structures.

The assessment of the antiproliferative effect of the tested polymers and nanoparticle formulation was further complemented through the determination of the cytoplasmic enzyme, LDH, released in the incubation medium, as a consequence of cell membrane disruption. A high amount of LDH released thus correlates with a high level of cell death. The highest concentrations tested in the previous assay (1000 and 2000 µg/mL) were those evaluated, and the obtained results are depicted in Figure 11.

A higher release of LDH was observed for FFv in both cell lines, supporting the MTT results and indicating a higher antiproliferative effect for the polymer when used by itself. When CS was tested, only the higher concentration (2 mg/mL) could generate an increased release of LDH, certainly an effect of the presence of acetic acid. With respect to the formulation of nanoparticles, an intermediate effect was observed, reflecting a contribution of both polymers. In both cell lines, the observations were mainly driven by the effect of time compared with concentration (*p* < 0.05). Importantly, the observations are generally in line with the results described above for the MTT assay. The behaviour of LDH release after exposure to fucoidan was studied elsewhere using HCT−15 cells [46]. The results show higher LDH release upon 24 h exposure to higher concentrations (100 µg/mL) of fucoidan from *Sargassum polycystim*, with IC_50_ found at 50 µg/mL [49]. A concentration- and time-driven behaviour was observed in a previous work of the team using fucoidan extracted from *Laminaria ochroleuca* [36]. Other works studied the effect on nanoparticles in a vaccine observing similar results [50]. Finally, another work used the same fucoidan (crude form, from *F. vesiculosus* from Sigma-Aldrich) and trimethylchitosan to obtain polyelectrolyte complexes, which were tested in A549 cells. No significant LDH release was reported for a concentration of 400 µg/mL, which was well below that tested in the present work [51].

The understanding of the antiproliferative effects of the tested polymers and nanoparticles allowed a better interpretation of the behaviour in living systems. In the first approach, the nanoparticle formulation showed lower antiproliferative activity when compared to the FFv alone. This occurrence was also noticed upon evaluation of the IC_50_. By evidencing decreased activity, higher concentrations of nanoparticles can be envisaged in future assays, although an adequate correlation to concentrations used in vivo must be assessed. Although the sulphated polysaccharide alone shows great potential against tumoral cells, the use of a fucoidan-based nanocarrier system could be proposed as a double-edged sword. In fact, combining the encapsulation of an anticancer agent with the antiproliferative effect herein discussed could be a therapeutic alternative or adjuvant in Schallenging anticancer treatments.

Despite the identified potential, the scarce knowledge of the toxicological profile, biodegradability, and pharmacokinetics of the involved materials [11,51,52,53,54] is a great limitation to further advancements. These aspects need to be further addressed before any formulation is truly considered.

## 3. Materials and Methods

### 3.1. Materials

Crude FFv, CS (low molecular weight, 75–85% deacetylation degree), gelatin from porcine skin (type A), barium chloride, trichloroacetic acid, 6-hydroxy-2,5,7,8-tetramethylchroman-2-carboxylic acid (Trolox), glacial acetic acid, Dulbecco’s Modified Eagle’s (DMEM), L-glutamine, non-essential amino acids, and 3-(4,5-dimethylthiazol-2-yl)-2,5-diphenyltetrazolium bromide (MTT) were purchased from Sigma-Aldrich (Burghausen, Germany). Potassium sulphate (K_2_SO_4_) and Folin-Ciocalteau reagent were from Panreac AppliChem (Barcelona, Spain). 2,2-azinobis(3-ethyl-benzothiazoline-6-sulfonate) (ABTS) was supplied by Alfa Aesar (Kandel, Germany), and sodium carbonate (Na_2_CO_3_) was purchased from Merck (Darmstadt, Germany). Phloroglucinol was supplied by AcrosOrganics (Shanghai, China). Polystyresulfonate sodium salt (PSS) standards were acquired from Sigma and Fluka (Barcelona, Spain and St. Louis, MO, USA, respectively). Phosphate-buffered saline (PBS) tablets pH 7.4 were acquired from VWR (Rosny-sous-Bois, France) and dimethyl sulfoxide (DMSO) from Carlo Erba (Sabadell, Spain). Penicillin/streptomycin and fetal bovine serum (FBS) were provided by Gibco (Life Technologies, Carlsbad, CA, USA), while the lactate dehydrogenase (LDH) kit was purchased from Takara Bio (Tokyo, Japan). Ultrapure water (Milli-Q, Millipore, Watford, UK) was used throughout. All other chemicals were reagent grade.

### 3.2. Cell Culture

Human colon colorectal carcinoma cells (HCT−116) and human lung epithelial adenocarcinoma cells (A549), both provided by the American Type Culture Collection (ATCC, Manassas, VA, USA), were used in passages 9–25 and 20–36, respectively. Cells were cultured in flasks (75 cm^2^), at 37 °C, in an incubator with 5% CO_2_/95% humidified atmospheric air. Cell culture media (CCM) was DMEM for both cell lines. For HCT−116 cells, DMEM was supplemented with penicillin/streptomycin at 1% (*v*/*v*) and FBS at 10% (*v*/*v*), while for A549 cells, CCM was supplemented with L-glutamine 200 mM (1%, *v*/*v*), non-essential amino acids (1%, *v*/*v*), penicillin/streptomycin at 1% (*v*/*v*), and FBS at 10% (*v*/*v*). CCM was exchanged two to three times weekly.

### 3.3. Characterisation of Fucoidans and Chitosan

#### 3.3.1. Oligosaccharide Content

The determination of the oligomeric fraction of fucoidan was performed in the aqueous phase. In brief, fucoidan was dissolved in ultrapure water and hydrolysed with H_2_SO_4_ (4%, *v*/*v*), for 20 min, at 121 °C and 2 atm. Afterwards, the solution was filtered (cellulose acetate, 0.45 μm, Sartorius, Göttingen, Germany) and analysed by High-Performance Liquid Chromatography (HPLC, 1100 Agilent, Waldbronn, Germany) using a refractive index detector (RI). The analysis was performed at 60 °C using an Aminex HPX-87H column (300 × 7.8 mm, BioRad, Hercules, CA, USA) and a mobile phase comprised of H_2_SO_4_ (0.003 M) at a flow rate of 0.6 mL/min. Data were obtained using fucose and galactose as standards. Furthermore, the results were expressed as O-Gal + Xyl + Man and O-Fucose (Sigma, Madrid, Spain), the former being in a group due to the inability of the column to separate the three saccharides.

#### 3.3.2. Antioxidant Properties

Antiradical properties were studied by the ABTS radical cation (ABTS^·+^) scavenging [55]. This determination was expressed as the trolox equivalent antioxidant capacity (TEAC) value, with trolox being the standard used to elaborate the calibration curve. For the determination, 1 mL diluted ABTS·+ solution was mixed with 10 µL of the sample (fucoidan), followed by an incubation period of 6 min, at 30 °C. Afterwards, the absorbances were read at 734 nm (*n* = 3), in a spectrophotometer (Evolution 201 UV-Vis, ThermoScientific, Waltham, MA, USA).

#### 3.3.3. Phenolic Content

Phenolic compounds were measured in fucoidan as phloroglucinol content [56]. Briefly, 1 mL of fucoidan solution was mixed with 1 mL of Folin-Ciocalteau reagent 1 N and 2 mL of sodium carbonate at 20% (*w*/*v*). After 45 min of incubation in darkness at room temperature, absorbances were measured (*n* = 3) by spectrophotometry (PharmaSpec UV−1700, UV-Visible spectrophotometer, Shimadzu, Hong Kong, China) at 730 nm. A standard curve was elaborated with phloroglucinol (*n* = 3).

#### 3.3.4. Molar Mass Distribution of the Polysaccharides

The molar mass distribution was estimated by High-Performance Size Exclusion Chromatography (HPSEC) using a modular system comprised of a Biotech Degasi GPC degasser, a Waters 515 HPLC pump, and a Knauer K-2300 refractive index detector. Two Shodex columns OHpak SB-806M HQ of 300 × 8.0 mm were used in series with a 50 × 6.0 mm OHpak SB-G 6B guard column.

A 0.2 M NaNO_3_ aqueous solution (0.02% NaN_3_) was used as mobile phase at 1 mL/min in the analysis of fucoidan, whilst for the analysis of chitosan it was a 0.2 M NaNO_3_ + 0.5 M acetic acid aqueous solution (0.02% NaN_3_), at the same rate. Fucoidan samples were dissolved in the eluent at 5 mg/mL and filtered through 0.45 µm filters. Chitosan samples were dissolved in the eluent at 5 mg/mL overnight, centrifuged at 4000 rpm for 10 min, and finally filtered through 0.45 µm filters. A conventional calibration curve was set using PSS standards dissolved at 5 mg/mL in the former eluent.

#### 3.3.5. Fourier Transform-Infrared Spectroscopy

FFv and CS were blended with KBr in a mortar and compressed into discs. Fourier Transform-Infrared Spectroscopy (FT-IR) spectra were collected by means of a 37-scan interferogram, in transmittance mode, in the 4000 to 400 cm^−1^ region, with a 4 cm^−1^ resolution (*n* ≥ 3, Bruker Tension 27, Bruker OPTIK GmbH, Ettlingen, Germany). An OPUS software (version 6.5) was used for data acquisition and analysis.

### 3.4. Thermophysical Features of Fucoidan, Chitosan, and Polymeric Nanoparticles: Analysis of Thermal Behaviour and Rheometry

#### 3.4.1. Differential Scanning Calorimetry

The thermal behaviour of the materials and nanoparticles (FFv/CS = 2/1) was evaluated using a differential scanning calorimeter (DSC, Q200, Netzsch, Selb, Germany). Each sample was weighed (5–15 mg) into aluminium pans, which were sealed, and their lids pierced. Measurements were performed between −80 °C and 400 °C, with various heating rates (10 °C/min and 20 °C/min), under a nitrogen atmosphere, and different cycles depending on the sample. Nitrogen was used as a purging gas, at a flow rate of 50 mL/min (*n* ≥ 2).

#### 3.4.2. Thermogravimetric Analysis

The thermal stability of all the materials and nanoparticles (FFv/CS = 2/1) was evaluated via thermogravimetric analysis (TGA 209-F1 Iris^®^, TA Instruments, Hüllhorst, Germany). The samples (up to 15 mg) were heated from 20 °C up to 1000 °C, at a heating rate of 10 °C/min, under a nitrogen atmosphere.

#### 3.4.3. Rheological Analysis

Steady-state shear measurements of tested polymers (FFv and CS) were performed at 25 °C on a controlled-stress rheometer (MCR 302, Paar Physica, Graz, Austria) equipped with a sand-blasted plate–plate measuring system (0.5 mm gap, 25 mm diameter). Flow curves of polymer dispersions (2%, *w*/*v*) were monitored over the shear rates range of 0.1 to 100 1/s. Flow curves of polymer dispersions (2%, *w*/*v*) were monitored over the shear rate range of 0.1 to 100 1/s. The biopolymer content was selected after preliminary steady-state shear measurements at different contents (from 0.5 to 5.0%), where no noticeable variations were identified in profile trends. The selected concentration was considered to provide a representative flow curve of the viscous behaviour of the tested polymers, while also being within the range commonly used in the food, pharmaceutical or cosmetic fields [57]. The thixotropy was also assessed by recording the forward and backward flow curves. Light silicon oil was used to seal the samples, which were rested in the measuring system for 10 min before rheological testing to enable the thermal and structural equilibration of the polymer solutions.

### 3.5. Production and Characterisation of Fucoidan/Chitosan Nanoparticles

#### 3.5.1. Production of Nanoparticles by Polyelectrolyte Complexation

To produce the nanoparticles, CS was dissolved in acetic acid 1% (*v*/*v*) to produce stock solutions at 1 and 10 mg/mL, which were filtered (5−13 µm, Whatman^®^, Dassel, Germany) before use. Fucoidan stock solution of FFv was also prepared at 1 and 10 mg/mL with ultrapure water and filtered (cellulose acetate, 0.45 µm, Frilabo, Milheirós, Portugal).

Nanoparticles were prepared through the slow addition of 0.8 mL of FFv onto 2 mL of CS at 1 mg/mL to obtain CS/fucoidan (CS/FFv) nanoparticles, or 0.8 mL of CS onto 2 mL of FFv at 1 mg/mL to obtain fucoidan/CS (FFv/CS) nanoparticles. The used volumes of FFv and CS were, thus, constant and only the concentrations of the added solutions varied in order to produce nanoparticles with different mass ratios (4/1; 3/1; 2/1; 1/1; 1/2; 1/3, and 1/4) for both CS/FFv and FFv/CS nanoparticles. These approaches allowed for elucidating the effect of the order of addition of the polymers and the use of different amounts of each polymer. In all cases, the formulations were prepared under gentle magnetic stirring at room temperature for 10 min.

The pH of the polymer solutions and nanoparticle dispersions was monitored throughout the process (PHS-25CW pH microprocessor, Scaninst, Hanna Instruments, Leighton Buzzard, UK). When adjustment of the pH of the polymer solutions was required, it was set at 4.0 and the adjustment was performed by the addition of 2–3 drops of hydrochloric acid or sodium hydroxide. A range of concentrations of the acid and base solutions was selected, with those being between 0.1 and 2 M.

#### 3.5.2. Characterisation of Nanoparticles

The morphology of the nanoparticles was observed by transmission electron microscopy (TEM) (JEM1010, 100 kV, JEOL, Tokyo, Japan). To do so, the samples were stained with 2% (*w*/*v*) phosphotungstic acid and placed on copper grids for TEM observation.

The nanoparticles were further characterised concerning their size, polydispersity index (PdI), and ζ-potential, using a Zetasizer Nano ZS (Malvern Panalytical, Malvern, UK). The characterisation was performed in an electrophoretic cell, after diluting 20 µL of the nanoparticle suspensions in 1 mL of ultrapure water (*n* ≥ 5).

The yield of nanoparticle production was determined by gravimetric analysis. Nanoparticles were prepared and isolated by centrifugation (16,000× *g*, 30 min, 15 °C). The solid fraction was freeze-dried (Labconco^®^ FreeZone 6 Liter Benchtop Freeze Dry System freeze dryer, Labconco, Kansas City, MO, USA) over 72 h (−60 °C, 2.37 × 10^−5^–3.95 × 10^–5^ atm). The production yield (PY) was calculated, as follows:PY (%) = (Nanoparticle weight/Total solids) × 100(1)
where nanoparticle weight refers to the nanoparticle sediment obtained after freeze-drying and the total solids is the total amount of solids added for nanoparticle production (*n* = 5).

The nanoparticle formulation FFv/CS = 2/1 (*w*/*w*) was also analysed by FT-IR, following the indications detailed in Section 2.3.

### 3.6. Antiproliferative Effect of Polymers and Nanoparticles

#### 3.6.1. Assessment of Metabolic Activity

The metabolic activity of HCT−116 and A549 cells after exposure to the raw materials FFv and CS, and to nanoparticles (FFv/CS = 2/1, *w*/*w*) was determined. Polymer samples were solubilised and nanoparticles were dispersed in CCM without FBS at several concentrations between 31.25 and 2000 µg/mL. The cells were seeded in 96-well plates at 1 × 10^4^ cells/well, incubated overnight, and exposed to the materials after that. CCM and SDS at 2% (*w*/*v*) were used as a negative and positive control of the antiproliferative effect, respectively. After 3, 24, and 48 h of exposure, the samples were removed and replaced by 30 µL of MTT (5 mg/mL in PBS pH 7.4), which remained under incubation for 2 h. After that, 50 µL of DMSO was added to each well to solubilise the formed formazan crystals. The absorbance was read by spectrophotometry (Infinite M200; Tecan, Grödig, Austria) at 540 nm, with background correction at 640 nm.

The cell viability was calculated, as follows:Cell viability (%) = [(A − D)/(CM − D)] × 100(2)
where A is the absorbance obtained upon exposure to each example, D represents the absorbance measured for DMSO and CM is the absorbance read for the cells incubated in CCM. The assay was performed at least 3 times, with 3 replicates for each tested concentration.

Using the obtained data, the half maximal inhibitory concentration (IC_50_) was calculated for those samples reaching less than 50% of cell viability in the conditions of the study. IC_50_ values were determined by sigmoidal fitting of the data in the GraphPad Prism statistical program (GraphPad Software, Version 6.01, USA).

#### 3.6.2. Assessment of Cell Membrane Integrity

Cell membrane integrity was assessed by the quantification of the cytoplasmic enzyme LDH on the supernatant of HCT−116 and A549 cells after 24 h and 48 h exposure to sample concentrations of 1000 and 2000 µg/mL. CCM was used as negative control and triton X−100 (1:100 dilution) as a positive control of LDH release. Supernatants were centrifuged (16,000× *g*, 5 min) and analysed with a commercial kit to quantify the released LDH. Absorbances were measured in a spectrophotometer (Infinite M200, Tecan, Grödig, Austria) at 490 nm and the background was corrected at 690 nm. The assay was carried out at least 3 times with 3 replicates for each tested concentration.

### 3.7. Statistical Analysis

The results were analysed by one-way analysis of variance (ANOVA) using Tukey’s multiple comparisons tests. The used software was GraphPad Prism^®^ (GraphPad software, Version 6.01, USA) and differences were considered significant at a level of *p* < 0.05.

## 4. Conclusions

This work provided an evaluation of the impact of the sulphate groups of crude commercial fucoidan from *F. vesiculosus* on nanoparticle production and cell antiproliferative effect. The obtained data evidenced potential for use as matrix material of polymeric nanocarriers, with a suggested impact on the proliferation of two cancer-derived cell lines (HCT−116 and A549 cells). The characterisation of the polymers was critical to understand the behaviour of the polymeric nanoparticles. The quantity of *F. vesiculosus* is correlated with the number of sulphate groups available to produce the polymeric nanoparticles by electrostatic interaction, and higher amounts of fucoidan result in smaller-sized nanoparticles (200–300 nm) displaying more negative zeta potential (>−40 mV). A production yield of 52% was determined for nanoparticles FFv/CS = 2/1 (*w*/*w*). Calorimetric data confirmed the stability of polymers involved in nanoparticle formulation at room temperature. Studies performed in carcinogenic cells demonstrated a time- and concentration-dependent antiproliferative effect of FFv, a behaviour that could be associated with the sulphate content. Nanoparticles evidenced a milder effect, certainly due to the lower availability of sulphate groups for interaction with cell structures. In future endeavours, the use of these polymers and nanoparticulate systems based on their composition can be considered for therapeutic approaches, contributing either with the antiproliferative capacity or with the matrix-forming ability. In any case, addressing toxicological concerns is imperative.

## Figures and Tables

**Figure 1 marinedrugs-21-00115-f001:**
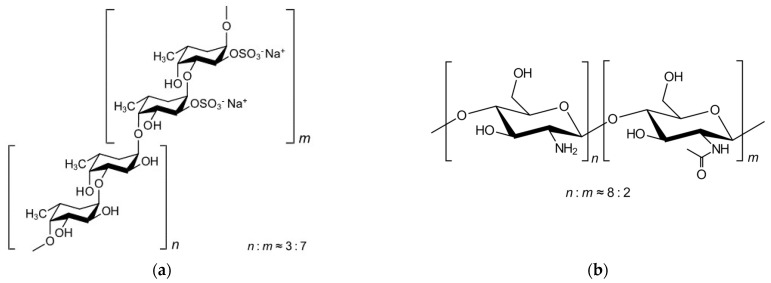
Chemical structures of (**a**) fucoidan from *F. vesiculosus* showing alternating α (1→3) and α (1→4) linkages [14] and (**b**) chitosan showing (1→4) bonds [15].

**Figure 2 marinedrugs-21-00115-f002:**
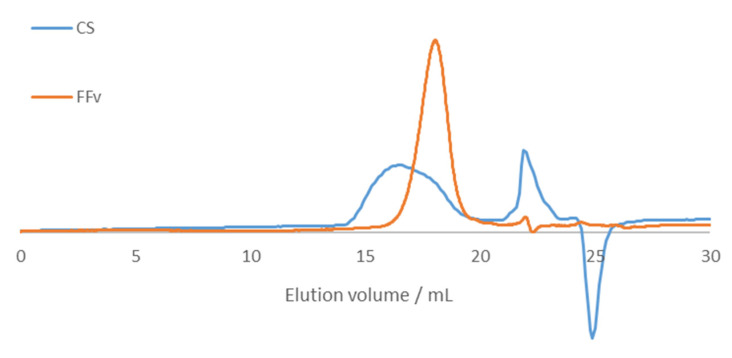
Size exclusion chromatography (SEC) elution profiles of crude fucoidan from the brown seaweed *F. vesiculosus* (FFv) and chitosan (CS).

**Figure 3 marinedrugs-21-00115-f003:**
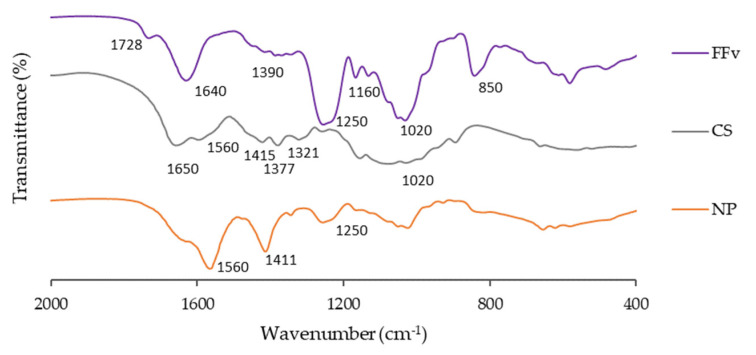
Fourier transform-infrared spectroscopy (FT-IR) of crude fucoidan from the brown seaweed *F. vesiculosus* (FFv), chitosan (CS), and a formulation of nanoparticles (NP; FFv/CS = 2/1, *w*/*w*).

**Figure 4 marinedrugs-21-00115-f004:**
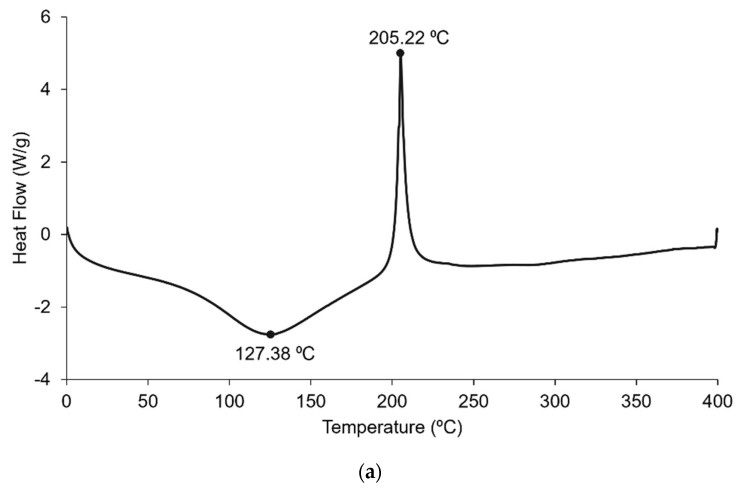
Differential scanning calorimetry (DSC) diagram obtained for (**a**) crude fucoidan from *F. vesiculosus*, (**b**) chitosan, and (**c**) nanoparticle formulation of FFv/CS = 2/1 (*w*/*w*).

**Figure 5 marinedrugs-21-00115-f005:**
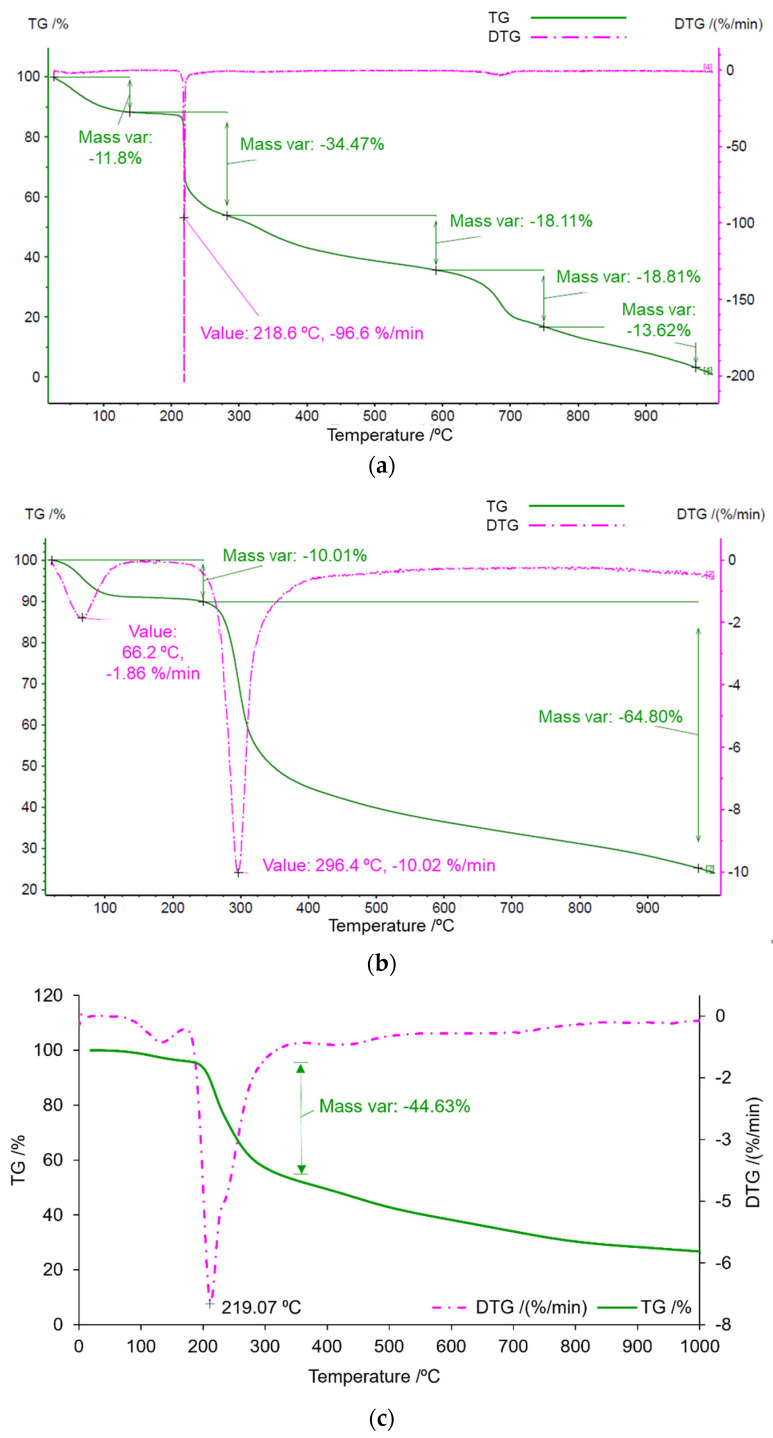
Thermogravimetric analysis diagram of (**a**) crude fucoidan from *F. vesiculosus* (FFv), (**b**) chitosan (CS), and (**c**) nanoparticle formulation of FFv/CS = 2/1 (*w*/*w*). Green (continuous) lines refer to the variation in the mass of the polymers throughout the process (TG/%), and pink (dashed) lines are the respective derivatives (DTG/(%/min)).

**Figure 6 marinedrugs-21-00115-f006:**
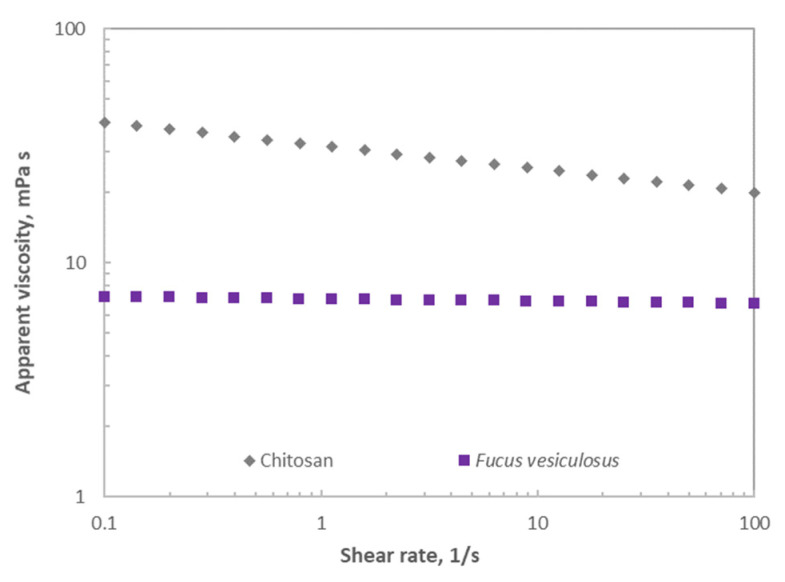
Flow curves for polymer solutions prepared with fucoidan from *F. vesiculosus* and chitosan.

**Figure 7 marinedrugs-21-00115-f007:**
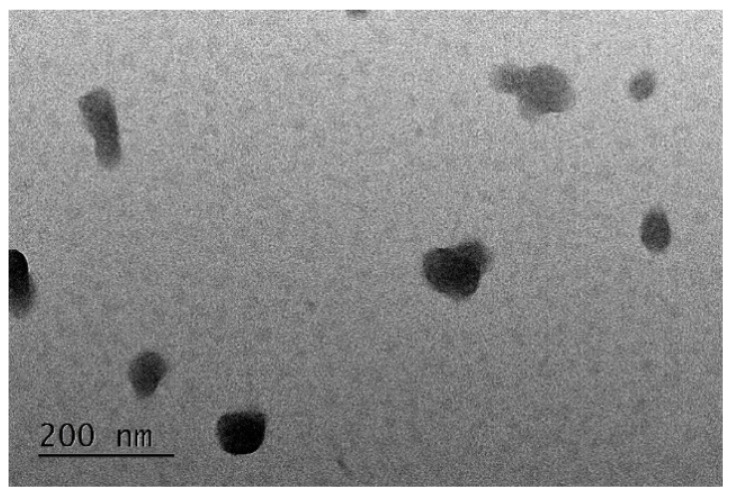
TEM microphotograph of FFv/CS = 2/1 (*w*/*w*) nanoparticles.

**Figure 8 marinedrugs-21-00115-f008:**
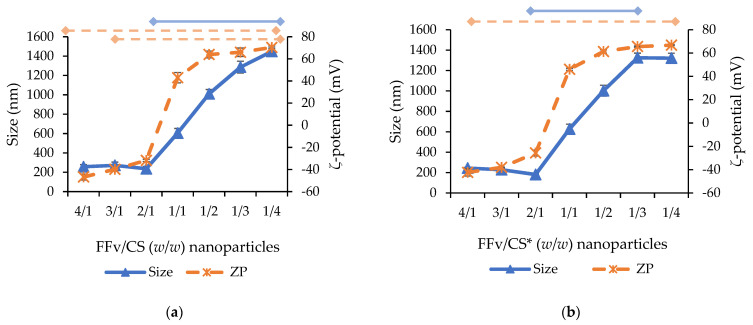
Physicochemical properties of nanoparticles formulated with fucoidan from *F. vesiculosus* (FFv) and chitosan (CS). FFv/CS indicates the addition of CS over FFv solution and CS/FFv indicates the addition of FFv over CS solution. The pH of polymer solutions was previously adjusted to 4 in (**b**,**d**) (pH adjustment = *), while (**a**,**c**) represented data of nanoparticles produced without pH adjustment. Data represent mean ± standard deviation (*n* ≥ 5). Top bars indicate statistical differences, a continuous line for size, and a dashed line for zeta potential (*p* < 0.05). Note: nanoparticles CS/FFv = 1/2 (*w*/*w*) resulted in precipitation.

**Figure 9 marinedrugs-21-00115-f009:**
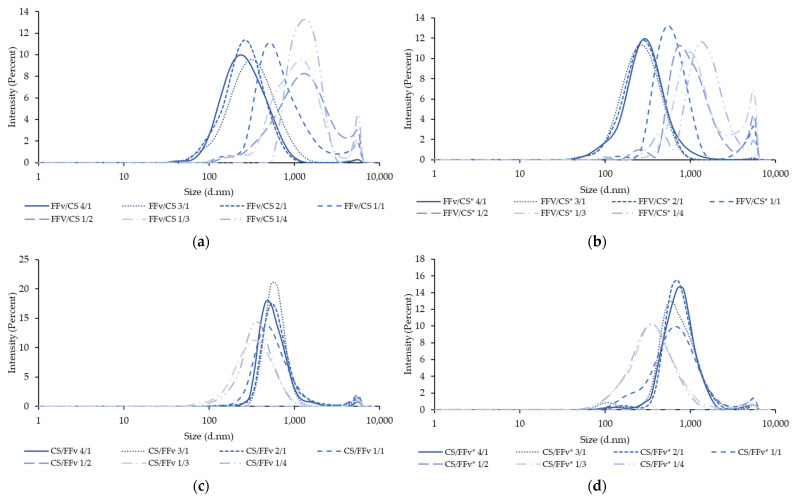
Size distribution by the intensity of nanoparticle formulations: (**a**) FFv/CS, (**b**) FFv/CS*, (**c**) CS/FFv, and (**d**) CS/FFv* at different ratios from 4/1 to 1/4 (*w*/*w*). CS: chitosan; FFv: fucoidan from *Fucus vesiculosus*; * pH adjustment.

**Figure 10 marinedrugs-21-00115-f010:**
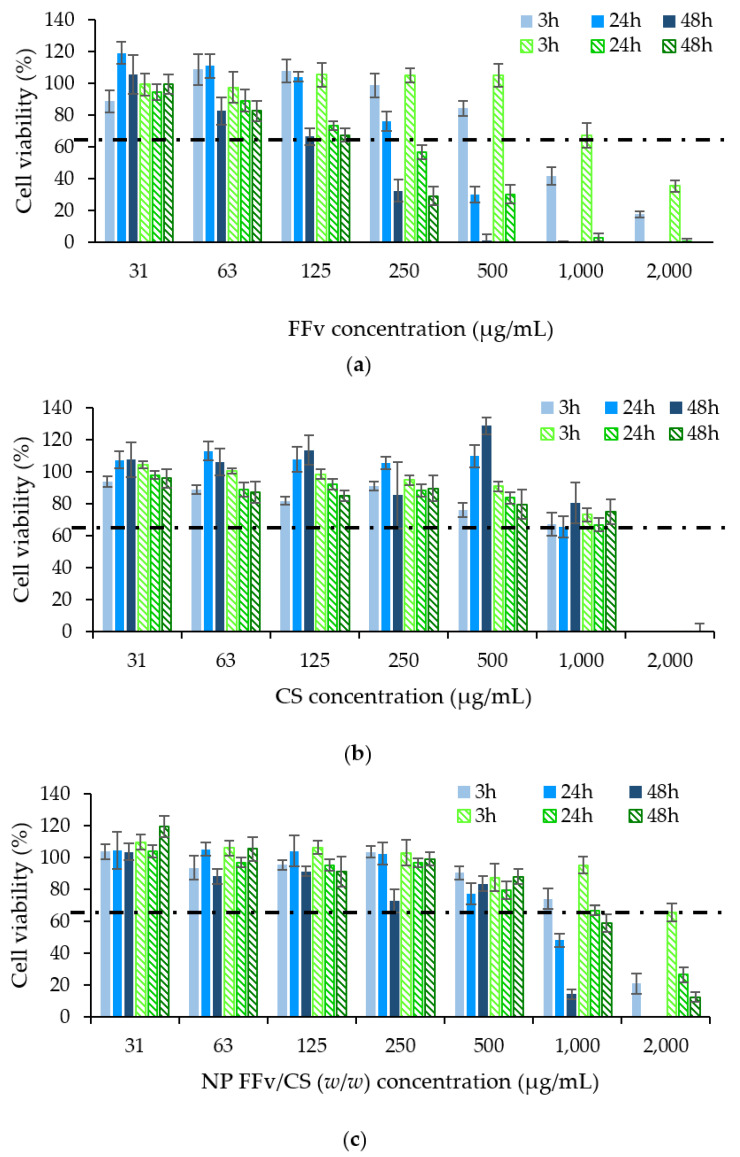
Cell viability induced by (**a**) fucoidan from *F. vesiculosus* (FFv), (**b**) chitosan, and (**c**) FFv/CS nanoparticles = 2/1 (*w*/*w*) upon 3 h, 24 h, and 48 h contact with HCT−116 (filled columns in blue: 

) and A549 (dashed columns in green: 

). Data represent mean ± SEM (*n* = 3). SEM: standard error of the mean.

**Figure 11 marinedrugs-21-00115-f011:**
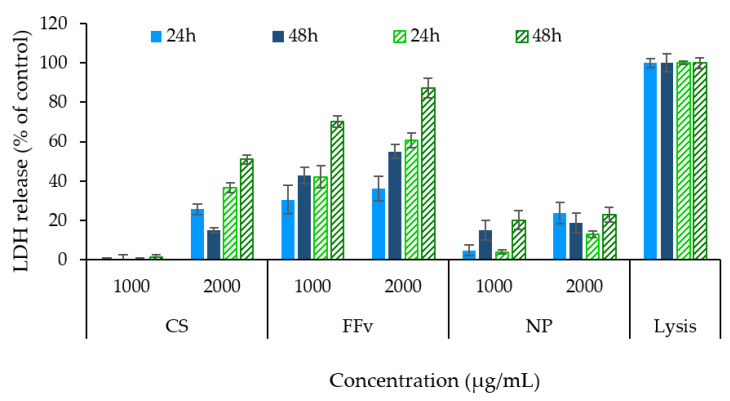
Lactate dehydrogenase release (LDH) from HCT−116 (filled columns in blue: 
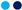
) and A549 cells (dashes columns in green: 
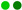
) upon contact with sample concentrations of 1 mg/mL and 2 mg/mL, at two different time points (24 h and 48 h). Data represent mean ± SEM (*n* = 3). SEM: standard error of the mean.

**Table 1 marinedrugs-21-00115-t001:** Characterisation of crude commercial fucoidan extracted from the brown seaweed *F. vesiculosus*.

Fucoidan from *F. vesiculosus*	% (*w*/*w*)
O-Xyl + Gal + Man	7.78 ± 0.17
O-Fucose	35.14 ± 1.12
Acetyl groups	9.68 ± 0.17
Phenolic content	2.65 ± 0.04
Sulphate content	27 *

O-: Oligosaccharide of; Xyl: xylose; Gal: galactose; Man: mannose; phenolic content as g phloroglucinol/100 g fucoidan; * calculated from the sulfur content provided by the supplier.

**Table 2 marinedrugs-21-00115-t002:** Calculated charge ratios in FFv/CS and Cs/FFv nanoparticles.

Nanoparticles FFv/CS (*w*/*w*)	1/4	1/3	1/2	1/1	2/1	3/1	4/1
±Charge ratio	0.15	0.20	0.30	0.60	1.19	1.79	2.38
Nanoparticles CS/FFv (*w*/*w*)	4/1	3/1	2/1	1/1	1/2	1/3	1/4

CS: chitosan; FFv: fucoidan from *Fucus vesiculosus.*

**Table 3 marinedrugs-21-00115-t003:** IC_50_ (μg/mL) calculated for fucoidan from *F. vesiculosus* (FFv) and nanoparticles (NP) FFv/CS = 2/1 (*w*/*w*).

Exposure Time	FFv		NP FFv/CS	
	HCT−116	A549	HCT−116	A549
3 h	928.1	1501.0	1344.0	n.a.
24 h	371.3	265.6	890.2	1256.0
48 h	165.6	160.7	674.8	1098.0

CS: chitosan; n.a.: not applicable.

## Data Availability

The data presented in this study are available on request from the corresponding author.

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
