# Peer review of "Fucoidan from Fucus vesiculosus: Evaluation of the Impact of the Sulphate Content on Nanoparticle Production and Cell Toxicity"

_marinedrugs, 2023, doi:10.3390/md21020115_

Round 1

Reviewer 1 Report (Previous Reviewer 2)

The authors have significantly improved the manuscript after the previous round of peer review. I have suggestions for additional consideration of the pharmacokinetics of chitosan (see, for example datails in  https://doi.org/10.1016/j.carbpol.2007.06.016;  https://doi.org/10.3390/md18110557)

Author Response

Reviewer 2 Report (Previous Reviewer 1)

Flórez-Fernández, et al. tried again to fulfill the previous comments raised for the production and characterization of FFv/CS nanoparticles and in vitro impact on cancerous cell lines. The authors have addressed most of the comments and the quality of the manuscript has been improved. The authors tried to do a correlation between the sulphate content in nanoparticles formulation and subsequent physicochemical and bioactivities. However, there are further comments that should be considered carefully:

-          What did the authors mean by sulphate groups in the title, is it sulphate groups content or pattern? I think they meant the sulphate content in the different nanoparticle formulations. If so, the title should be revised again.

-          The new title reflected the magic effect of the fucoidan’s sulphate ester groups. Does it mean that other sulphated polysaccharides cannot do the same or similar functions?

-          It was clear that the sulphate groups content affected the physicochemical characters of formulation (e.g., Tyndall effect and zeta potential), and therefore, on the bioactivities afterwards (the milder effect due to availability of sulphate groups to cancerous cell lines). Hence, the sulphate content should be also determined in the different formulations to obtain a realistic correlation. Otherwise, the bioactivities of the different formulations should be also evaluated.

-          Other characters other than zeta potential should be investigated to reveal the effect of sulphate content and availability. The sulphate groups are always ionized at all pH values. So, the effect of pH on zeta potential is not clear in Figure 8.

-         -          What is the importance of the TEAC abbreviation mentioned at Table 1’s legend?

-          -          It is still not clear why the authors choose the 2% (w/v) to investigate the rheological analysis.

-          Are there any other TEM images showing more uniform nanoparticles? Also, the nanoparticles sizes stated 200-300 nm in the abstract, while they were below 200 nm as stated in the manuscript and Figure 7.

-          The title of phloroglucinol content should be revised to phenolic content. The unit also should be … phloroglucinol equivalent/.. fucoidan

-          Why was the antioxidant capacity determined? If there is a relationship between antioxidant and antiproliferative effect, it should be clarified, and the results should be not only correlated with fucoidan sulphate groups availability. The antioxidant activity represented by ABTS radical scavenging was not discussed within the Results and discission section.

-          The meaning of SEM abbreviation in Figure 10 and 11 legends should be clarified.

Conclusively, the manuscript may be revised to be clearer and more informative for the readers. Several aspects were discussed, but they should be correlated.

Round 2

Reviewer 2 Report (Previous Reviewer 1)

-          The sulphate content of the different formulations should be determined. The authors replied that it is difficult because they are not sure about the NP solubility. Then, elemental analysis is enough for this purpose and doesn’t need any kind of solubilization,

-          The relation between antioxidant activity and cytotoxic activity should be clarified. The antioxidant activity is related more to phenolic content rather than to sulphate content which is the main topic of this research.

Author Response

The authors thank the reviewer for the time and effort in reviewing once again the new revised version of the manuscript. 

Reply to the reviewer’ questions:

  • The sulphate content of the different formulations should be determined. The authors replied that it is difficult because they are not sure about the NP solubility. Then, elemental analysis is enough for this purpose and doesn’t need any kind of solubilization,

The authors recognise the point made by the reviewer. Nevertheless, we should go back to the answer provided in the previous round of comments. Nanoparticles produced with different mass ratios of the materials can be prepared and characterised, as the technique for characterisation (light scattering) is highly sensitive and is not disturbed by the free polymer. However, other studies like the one of elemental analysis or other possible methods for the determination of sulphate content require a minimum amount of the nanoparticles that need to be isolated (related with the production yield, also referred to in the manuscript). Therefore, with the exception of formulation 2/1, it would not be possible to obtain the required amount, hindering any comparison.

The rational of our work was to characterise the polymers, namely FFv, which is less known for the community, described and characterised. The molecular formula of the used natural polymers is just approximate and so we prepare the nanoparticles with varied compositions aiming at finding the best combination, where higher interactions take place. If the molecular formulas of these kind of polymers was precisely known, a simple calculation on defined combinations would permit determining the best conditions. Therefore, with the approach of testing different compositions (polymer mass ratios), we intend to find the most approximate to equimolar concentrations without observing precipitation, which is expected to show the higher yield. After obtaining the best formulation, the nanoparticles were used for further studies, as the biological activity determination. 

2) The relation between antioxidant activity and cytotoxic activity should be clarified. The antioxidant activity is related more to phenolic content rather than to sulphate content which is the main topic of this research.

This comment of the reviewer has two inherent questions:

  1. Relation between antioxidant activity and cytotoxic activity should be clarified

In this work, the antioxidant activity was measured only by TEAC assay. The authors follow the last reply about this question: “Other works have reported effects of antioxidant properties on antiproliferative activity, but the simple determination of TEAC value for FFv does not allow a direct correlation.”

Please, let us know if you are aware of a work including this information. Then, we could contrast and amend our manuscript.

  1. The antioxidant activity is related more to phenolic content rather than to sulphate content with is the main topic of this research.

Following your comment, the authors revised the bibliography to clarify this issue. In a comprehensive review entitled: “Biological Activities of Fucoidan and the Factors Mediating Its Therapeutic Effects: A Review of Recent Studies” published in Marine Drugs in 2019 (doi: 10.3390/md17030183) it can be read in the section about antioxidant activity “A positive correlation between sulfate content and antioxidant capability has been reported. Moreover, the ratio of sulfate content and fucose influences hydroxyl radical scavenging ability [4,5].” Besides, in this work was also written: “Most importantly, the substituents of fucoidan play a major role in its antioxidant activity [62,67]. Wang et al. studied the antioxidant mechanisms of LMWF and identified an influence of the substituent groups [67]”, LMWF being low molecular weight fucoidan.

According to the aforementioned, the statement that the antioxidant activity is more related to phenolic content than to sulphate content, cannot be assumed.

This manuscript is a resubmission of an earlier submission. The following is a list of the peer review reports and author responses from that submission.

Round 1

Reviewer 1 Report

The current manuscript presented by Flórez-Fernández, et al. investigated the pH dependent formulation and characterization of fucoidan/chitosan nanoparticles (NPs). However, the research novelty is low, since a lot of previously published reports have addressed these kinds of experiments which were reviewed recently in a book chapter by Zayed, et al. 2022 (https://doi.org/10.1016/B978-0-12-822351-2.00017-6). In addition, the optimum formulation failed to improve fucoidan potency against cancerous cell proliferation. Moreover, the use of fucoidan/chitosan NPs formulation as carriers for drug delivery system has been intensively investigated. Hence, the present manuscript suffers from various defects, including:

-    The methodology involved the use of fucoidan derived from F. vesiculosus purchased from Sigma-Aldrich, which product was used, since Sigma produced two types of this fucoidan, i.e., crude and pure form?

-          The last point should be clarified, since the phenolic content can interfere with the investigated bioactivity,

-          Comparison with previous publications and reference bench markers was also missed in biological studies,

-          How was the pH adjusted during the NPs formulation? Was it just only monitoring by pH meter as addition of either acid or alkali or working in a specific buffer?

Formulation needs also revision regarding,

-          FTIR (Figure 3) of NPs is not well discussed to confirm the real formation of NPs composed of Fucoidan and chitosan,

-          "Nanoparticles of 200-300 nm were obtained when the fucoidan amount was prevalent." as mentioned in the abstract, TEM or SEM is also needed to visualize the formed polymeric NPs. Nevertheless, nothing was mentioned in experimental part,

-          DSC diagrams for fucoidan and chitosan were shown respectively, did the authors do DSC for NPs as well to investigate the thermal behavior after the addition of fucoidan to chitosan? (NPs dispersion elucidated after polyelectrolyte complexation could be easily lyophilized to be ready for DSC). The same might be considered for TGA.

-          The authors demonstrated the rheological analysis of the polymeric solution. However, why did the authors select 2 %w/v, since the rheological behavior might be concentration dependent (mainly for CS)?,

-          Different mass ratios were used during the NPs formulation. So, it is preferable to use the same concentration employed in the formulation and test the rheology using physical mixing between both polymers to investigate the effect of this combination on the viscosity.

Author Response

The current manuscript presented by Flórez-Fernández, et al. investigated the pH dependent formulation and characterization of fucoidan/chitosan nanoparticles (NPs). However, the research novelty is low, since a lot of previously published reports have addressed these kinds of experiments which were reviewed recently in a book chapter by Zayed, et al. 2022 (https://doi.org/10.1016/B978-0-12-822351-2.00017-6). In addition, the optimum formulation failed to improve fucoidan potency against cancerous cell proliferation. Moreover, the use of fucoidan/chitosan NPs formulation as carriers for drug delivery system has been intensively investigated. Hence, the present manuscript suffers from various defects, including:

- The methodology involved the use of fucoidan derived from F. vesiculosus purchased from Sigma-Aldrich, which product was used, since Sigma produced two types of this fucoidan, i.e., crude and pure form?

We thank the reviewer for this comment. The used fucoidan was crude, according to the supplier information. This information was now added to the manuscript, please see new manuscript version, line 402, reading: “FFv crude was purchased from Sigma-Aldrich (Germany).”

- The last point should be clarified, since the phenolic content can interfere with the investigated bioactivity

The reviewer is correct when saying that phenolic contents will be different between crude and pure fucoidan. It is also correct that the presence of reducing agents in the samples being analysed will affect the results of MTT-like assays, which are based on a reducing reaction. Nevertheless, in the present work that is not a concern because the protocol used to evaluate the cell viability includes a step where the samples are removed from their contact with cells before the MTT reactive is added. That was already indicated in the description of the method: “The cells were seeded in 96-well plates at 1·104 cells/well, incubated overnight, and exposed to the materials after that. CCM and SDS at 2% (w/v) were used as negative and positive control of the antiproliferative effect, respectively. After 3, 24 and 48 h of exposure, the samples were removed and replaced by 30 µL of MTT (5 mg/mL in PBS pH 7.4), which remained under incubation for 2 h.”

- Comparison with previous publications and reference bench markers was also missed in biological studies

Section 2.3. of the manuscript refers to the biological studies performed in this work, which refer to the antiproliferative effect of polymers and nanoparticles, where “Two epithelial cell lines, HCT-116 and A549, from intestinal and lung epithelia, respectively, were used to better understand the effects on proliferation upon exposure to CS, FFv and the nanoparticle formulation FFv/CS = 2/1 (w/w).” Following the comment of the reviewer, the text was restructured to include more comparisons with the literature. Please, see lines 356 to 365: “Another work described the growth inhibition of 50% for A549 cells upon exposure to 300 µg/mL of the same FFv (crude fucoidan from F. vesiculosus from Sigma) at a timepoint of 48 h, a value around 2-fold higher than the IC50 obtained herein. This observation reveals lower toxicity determined in that work, when compared to the present work, in which less fucoidan is needed to inhibit cell proliferation [39]. A concentration of 2000 µg/mL of fucoidan extracted from Kelp (family: Laminariaceae) was also demonstrated to inhibit HCT-116 cell viability after 24 h reaching cell viability close to IC50 [44]. These observations suggest that, when fucoidan is used alone, the sulphate groups of the polymer are more available to mediate interactions with cellular structures, having stronger impact on cell viability.”; and lines 386 to 396: “Importantly, the observations are generally in line with the results described above for the MTT assay. The behaviour of LDH release after exposure to fucoidan was studied elsewhere using HCT-15 cells [45]. The results show higher LDH release upon 24 h exposure to higher concentrations (100 µg/mL) of fucoidan from Sargassum polycystim, being the IC50 found at 50 µg/mL [45]. A concentration- and time-driven behaviour was observed in a previous work of the team using fucoidan extracted from Laminaria ochroleuca [33]. Finally, another work used the same fucoidan (crude form, from F. vesiculosus from Sigma-Aldrich) and trimethylchitosan to obtain polyelectrolyte complexes, which were tested in A549 cells. No significant LDH release was reported for a concentration of 400 µg/mL, which is well below that tested in the present work [46].”

- How was the pH adjusted during the NPs formulation? Was it just only monitoring by pH meter as addition of either acid or alkali or working in a specific buffer?

The pH adjustment was performed using HCl and NaOH, 2 or 3 drops to avoid modifying the concentration of the polymers (fucoidan or chitosan). Different concentrations of HCl and NaOH were used, 0.1 M to 2 M according to the needs. Following the reviewer comment, the text was clarified, please see on lines 532 to 536: “The pH of the polymer solutions and nanoparticle dispersions was monitored throughout the process (PHS-25CW pH microprocessor, Scaninst, Hanna Instruments, UK). The optimal working pH was set at 4.0 and its adjustment was performed using 2-3 drops of hydrochloric acid or sodium hydroxide at variable concentrations from 0.1 M to 2 M, which were added to the polymer stock solutions.”

Formulation needs also revision regarding, FTIR (Figure 3) of NPs is not well discussed to confirm the real formation of NPs composed of Fucoidan and chitosan,

Thank you for your comment, the authors revised the discussion of Figure 3. Please see the new version of the manuscript, lines 155 to 170: “The formulation of the NP using CS and FFv at 2/1 ratio (w/w) was also characterized. The corresponding spectrum is clearly dominated by the bands of CS, thus reflecting its composition. The C=O of FFV is now masked by the broad 1650-1560 cm-1 band of CS. The latter absorption is now the most intense, which seems to corroborate the attribution of the 1640 cm-1 band to adsorbed water: an effective drying of the formulation would eliminate part of this water. Also, during the polyelectrolyte complexation, solvation water molecules are expelled from the coacervate, thus reducing the amount of adsorbed water in the formulation relative to the free polyelectrolytes. The band centered at 1020 cm-1, associated to the glycosidic linkage, also reflects the presence of both polymers, since it is as broad as that of CS, but not as round-shaped; instead, it presents a sharper shape, similar to the one of FFv. Lastly, the band attributed to the protonated amino groups of CS shifted to 1411cm-1 and became broader, now masking the C-H bendings of acetyl groups. This may be ascribed to the electrostatic binding of such groups and the sulfates of FFV. These observations were expected, demonstrating the complexation between the polymers resulting, ultimately, in the polymeric nanoparticles. Other authors, formulating nanoparticles based on the polymers focused herein, observed similar performance [16,25].” 

- "Nanoparticles of 200-300 nm were obtained when the fucoidan amount was prevalent." as mentioned in the abstract, TEM or SEM is also needed to visualize the formed polymeric NPs. Nevertheless, nothing was mentioned in experimental part.

The authors thank the reviewer for the suggestion but, regrettably, don’t have the possibility to perform this assay as the fucoidan of the batch used to prepare the nanoparticles is over. Nevertheless, dynamic light scattering is the technique most adequate to determine the size of nanoparticles. Microscopy techniques such as TEM or SEM would be very useful to determine nanoparticles morphology but not their size, because the required application of vacuum on the samples during visualization tends to result in the underestimation of their size, a characteristic we wanted to evaluate rigorously. In any case, we will consider this experimental characterisation in future works.

- DSC diagrams for fucoidan and chitosan were shown respectively, did the authors do DSC for NPs as well to investigate the thermal behavior after the addition of fucoidan to chitosan? (NPs dispersion elucidated after polyelectrolyte complexation could be easily lyophilized to be ready for DSC). The same might be considered for TGA.

The authors appreciate the suggestion of the reviewer. This experimental section was focused solely on the characterisation of the polymers. Moreover, nanoparticle formation is mediated by the interaction between opposite charges of the polymers and, thus, no difference is expected when comparing the polymers alone and the nanoparticles. Furthermore, both polymers are amorphous, showing a very similar thermal behaviour, only differing in the degradation temperature. The latter is not important, at this point and for this work, as the temperature at which the nanoparticles were produced is mild. Therefore, considering all the information and aspects, we consider that there is no real necessity to analyse these thermal behaviours for the nanoparticles. However, in future works, the same experimental rationale will be thought of, and its feasibility considered.

- The authors demonstrated the rheological analysis of the polymeric solution. However, why did the authors select 2 %w/v, since the rheological behavior might be concentration dependent (mainly for CS)?

The reviewer is right that the rheological behaviour of a polymeric solution is dependent on several factors such as concentration, temperature, presence of bubbles, shear rate, among many others. Therefore, an exhaustive analysis of the rheological behaviour would be the object of an independent study and would be outside the subject of this work. Anyway, we performed preliminary steady-state shear measurements at different concentrations in order to select one flow curve representative of the viscous behaviour of the tested biopolymers, and also within the ranges of biopolymer commonly used in the food, pharmaceutical or cosmetic fields. It was clarified, please see lines 509-510: “The selected biopolymer concentration is within the range commonly used for natural materials in the food, pharmaceutical or cosmetic fields [49].”

- Different mass ratios were used during the NPs formulation. So, it is preferable to use the same concentration employed in the formulation and test the rheology using physical mixing between both polymers to investigate the effect of this combination on the viscosity.

Thank you for drawing our attention on this issue. In this work, the viscosity measurements were only focused on the polymers themselves used as a base, but we will keep it in consideration for future works.

We thank the reviewer for the comments and suggestions to improve our manuscript to be published in Marine Drugs.

Reviewer 2 Report

Noelia Flórez-Fernández et al. have studied the t characterized commercial fucoidan obtained from Fucus  vesiculosus and used it for the preparation of nanoparticles. AAfter close evaluation of the manuscript I have following comments and recommendations.

1. In Abstract: The conclusion "hese marine materials (fucoidan and chitosan) provided features suitable to formulate polymeric nanoparticles to use in biomedical applications  as drug carrier." is too ambigous. Authors have not proved nanoparticles as drug careers.

2. Beside potential career function, fucoidan have own pharmacological effects. These aspects should be addressed in Introduction

3.. All abbreviations in Table 1 should be explained.

4. Sect. 2.1.1: Beside extraction, purification and fractionation methodologies , the collection place, reproductive phase could also affect the biochemical composition and biological properties of fucoidan ( https://doi.org/10.3390/md20030193).

5. Fig. 6: why rheological properties for 2%, w/v solutions only were studied?

6. In Sect 2.2.6: How uniform nanoparticles were , The particle distribution diagrams would be of interest.

7. In Sect. 2.3: In case nanoparticles are planned for drug delivery, the pharmacokinetic of compounds forming nanoparticles (fucoidan and chitosan) is very important. Please address this question.

Author Response

Noelia Flórez-Fernández et al. have studied the characterized commercial fucoidan obtained from Fucus vesiculosus and used it for the preparation of nanoparticles. After close evaluation of the manuscript I have following comments and recommendations.

  1. In Abstract: The conclusion "these marine materials (fucoidan and chitosan) provided features suitable to formulate polymeric nanoparticles to use in biomedical applications as drug carrier." is too ambigous. Authors have not proved nanoparticles as drug careers.

We thank the reviewer for the comment and understand the concern. The text was modified, please see in the abstract: "These marine materials (fucoidan and chitosan) provided features suitable to formulate polymeric nanoparticles to use in biomedical applications."

  1. Beside potential career function, fucoidan have own pharmacological effects. These aspects should be addressed in Introduction

Following the reviewer comment, the introduction was modified, please see lines 50 to 55: “Whereas its specific structure may be highly variable, depending on the source, the structural position of sulphate groups has been correlated with the resulting biological activity, including antioxidant and antitumoral, as well as with the applications, including in drug delivery [4,5]. Furthermore, other possible therapeutic approaches can be envisaged, mainly focusing on the pharmacological potential of fucoidan that has been described elsewhere [6–8].”

  1. All abbreviations in Table 1 should be explained.

Thank you for the advice, Table 1 was modified, please see lines 99-102.

  1. Sect. 2.1.1: Beside extraction, purification and fractionation methodologies, the collection place, reproductive phase could also affect the biochemical composition and biological properties of fucoidan (https://doi.org/10.3390/md20030193).

The text was modified for completeness of information. The proposed reference was added, please see lines 89-91: “The indication that the composition of sulphated polysaccharides extracted from al-ga is influenced by different factors mainly related with environmental parameters and the extraction method is solid [13,14].”

  1. Fig. 6: why rheological properties for 2%, w/v solutions only were studied?

This comment was also done by the other reviewer. The reviewer is right that the rheological behaviour of a polymeric solution is dependent on several factors such as concentration, temperature, presence of bubbles, shear rate, among many others. Therefore, an exhaustive analysis of the rheological behaviour would be the object of an independent study and would be outside the subject of this work. Anyway, we performed preliminary steady-state shear measurements at different concentrations in order to select one flow curve representative of the viscous behaviour of the tested biopolymers, and also within the ranges of biopolymer commonly used in the food, pharmaceutical or cosmetic fields. Please see lines 509-510: “The selected biopolymer concentration is within the range commonly used for natural materials in the food, pharmaceutical or cosmetic fields [49].”

  1. In Sect 2.2.6: How uniform nanoparticles were, the particle distribution diagrams would be of interest.

We thank the reviewer for this question. The characterisation of nanoparticle size resulted in a measurement of size, but also of the polydispersity index (PdI), which mistakenly was not mentioned in the discussion. PdI values below 0.2 indicate great homogeneity but are very difficult to attain, particularly when natural polymers are used. In the case of our formulations, lower PdIs were always observed when fucoidan was predominant. All in all, the PdI varied between 0.2 and 0.4, which is considered acceptable. A comment on this parameter was added to the text, please see lines 280 to 284, that read as follows: " An important feature concerning the size of the nanoparticles is the polydispersity index (PdI), which is indicative of homogeneity. Both FFv/CS and CS/FFv nanoparticles dis-played PdI between 0.2 and 0.4, which is considered acceptable, particularly because natural polymers are being used.”

  1. In Sect. 2.3: In case nanoparticles are planned for drug delivery, the pharmacokinetic of compounds forming nanoparticles (fucoidan and chitosan) is very important. Please address this question.

The authors thank the reviewer for this comment, as it further delves into the characterization and behaviour of these polymers. The (scarce) literature refers to the pharmacokinetic profile of fucoidan as mainly having a renal excretion. After oral administration of the polymer, several authors report that it has a faster distribution from blood to tissues, only to be found in increased concentrations in the kidneys, and in moderate amounts in the liver and in the spleen (Wang et al., 2021 - 10.1039/d0fo03153d; Bai et al., 2020 - 10.3390/molecules25051087).

As for chitosan, the literature refers to its pharmacokinetic profile in dependence of the used derivative, if more water soluble or insoluble, and its molecular weight. Despite that, chitosan can be absorbed via intestinal epithelial cells, and it is reported an accumulation in several organs, such as the liver, kidneys and spleen, apart from presence in blood. The concentrations reported for blood are lower than those found in the liver, revealing a possible accumulation in the latter. Hepatic enzymes are the ones responsible for degrading chitosan, and also lysozyme, widely present in mucosal tissues (Wu et al., 2017 - 10.1016/j.carbpol.2016.08.076), and its metabolites are further excreted in the urine.

Further research needs to be conducted to assess this question, naturally.

The authors considered that it can be helpful for the discussion of the overall results the inclusion of a comment pertaining to this question, which was inserted in lines 408 to 411, reading: “Despite the identified potential, the scarce knowledge of the toxicological profile, biodegradability and pharmacokinetics of the involved materials is a great limitation to further advancements. These aspects need to be further addressed before any formulation is truly considered.”.

We thank the reviewer for the suggestions and comments, which contributed to strongly improve our article proposal. 

Round 2

Reviewer 1 Report

Unfortunately, the criticisms of the first expert report were only partially remedied. In particular, the comments on the experimental implementation were not taken into account. It was pointed out that the experiments should be completed and the comments included in later experiments. Therefore, the weak points unfortunately remain. Therefore, the manuscript should not be accepted for publication, but a new paper should be submitted after further experiments have been conducted.   

Reviewer 2 Report

Authors have revised and updated the manuscript. However some questions require additional improvement.

  1.  In Sect 2.2.6: How uniform nanoparticles were, the particle distribution diagrams would be of interest.
I believe that particle distribution diagrams  will be suitable for illustration (at least in supplementary materials).

 2.   In Sect. 2.3: In case nanoparticles are planned for drug delivery, the pharmacokinetic of compounds forming nanoparticles (fucoidan and chitosan) is very important. Please address this question.
I suggest to discuss next data which provide a lot of important information:
https://doi.org/10.3390/md16040132;
https://doi.org/10.3390/md17120687;
https://doi.org/10.3390/md16040132